# MINING PATENTS WITH LARGE LANGUAGE MODELS ELUCIDATES THE CHEMICAL FUNCTION LANDSCAPE

## ABSTRACT

The fundamental goal of small molecule discovery is to generate chemicals with target functionality. While this often proceeds through structure-based methods, we set out to investigate the practicality of orthogonal methods that leverage the extensive corpus of chemical literature. We hypothesize that a sufficiently large text-derived chemical function dataset would mirror the actual landscape of chemical functionality. Such a landscape would implicitly capture complex physical and biological interactions given that chemical function arises from both a molecule's structure and its interacting partners. To evaluate this hypothesis, we built a Chemical Function (CheF) dataset of patent-derived functional labels. This dataset, comprising 631K molecule-function pairs, was created using an LLM- and embedding-based method to obtain functional labels for approximately 100K molecules from their corresponding 188K unique patents. We carry out a series of analyses demonstrating that the CheF dataset contains a semantically coherent textual representation of the functional landscape congruent with chemical structural relationships, thus approximating the actual chemical function landscape. We then demonstrate that this text-based functional landscape can be leveraged to identify drugs with target functionality using a model able to predict functional profiles from structure alone. We believe that functional label-guided molecular discovery may serve as an orthogonal approach to traditional structure-based methods in the pursuit of designing novel functional molecules.

## 1 INTRODUCTION

The overarching goal of drug discovery is to generate chemicals with specific functionality through the design of chemical structure (Li & Kang, 2020). Functionality, often in the context of drug discovery, refers to the specific effects a chemical exhibits on biological systems (i.e., vasodilator, analgesic, protease inhibitor), but it is applicable to materials as well (i.e., electroluminescent, polymer). Computational methods often approach molecular discovery through structural and empirical methods such as protein-ligand docking, receptor binding affinity prediction, and pharmacophore design (Corso et al., 2022; Trott & Olson, 2010; Wu et al., 2018; Yang, 2010). These methods are powerful for designing molecules that bind to specific protein targets, but at present they are unable to explicitly design for specific organism-wide effects. This is largely because biological complexity increases with scale, and many whole-body effects are only weakly associated with specific protein inhibition or biomolecular treatment (Drachman, 2014).

Humans have long been documenting chemicals and their effects, and it is reasonable to assume functional relationships are embedded in language itself. Text-based functional analysis has been paramount for our understanding of the genome through Gene Ontology terms (Consortium, 2004). Despite its potential, text-based functional analysis for chemicals has been largely underexplored. This is in part due to the lack of high-quality chemical function datasets but is more fundamentally due to the high multi-functionality of molecules, which is less problematic for genes and proteins. High-quality chemical function datasets have been challenging to generate due to the sparsity and irregularity of functional information in chemical descriptions, patents, and literature. Recent efforts at creating such datasets tend to involve consolidation of existing curated descriptive datasets (Wishart et al., 2023; Degtyarenko et al., 2007). Similarly, keyword-based function extraction partially solves the function extraction problem by confining its scope to singular predetermined functionality, but it fails at broadly extracting all relevant functions for a given molecule (Subramanian

et al., 2023). Given their profound success in text summarization, Large Language Models (LLMs) may be ideal candidates to broadly extract functional information of molecules from patents and literature, a task that remains unsolved (Brown et al., 2020; OpenAI, 2023; Touvron et al., 2023). This is especially promising for making use of the chemical patent literature, an abundant and highly specific source of implicit chemical knowledge that has been largely inaccessible due to excessive legal terminology (Senger, 2017; Ashenden et al., 2017). This may allow for the creation of a large-scale dataset that effectively captures the text-based chemical function landscape.

We hypothesize that a sufficiently large chemical function dataset would contain a text-based chemical function landscape congruent with chemical structure space, effectively approximating the actual chemical function landscape. Such a landscape would implicitly capture complex physical and biological interactions given that chemical function arises from both a molecule's structure and its interacting partners (Martin et al., 2002). This hypothesis is further based on the observation that function is reported frequently enough in patents and scientific articles for most functional relationships to be contained in the corpus of chemical literature (Papadatos et al., 2016). To evaluate this hypothesis, we set out to create a Chemical Function (CheF) dataset of patent-derived functional labels. This dataset, comprising 631K molecule-function pairs, was created using an LLM- and embedding-based method to obtain functional labels for approximately 100K molecules from their corresponding 188K unique patents. The CheF dataset was found to be of high quality, demonstrating the effectiveness of LLMs for extracting functional information from chemical patents despite not being explicitly trained to do so. Using this dataset, we carry out a series of experiments alluding to the notion that the CheF dataset contains a text-based functional landscape that simulates the actual chemical function landscape due to its congruence with chemical structure space. We then demonstrate that this text-based functional landscape can be harnessed to identify drugs with target functionality using a model able to predict functional profiles from structure alone. We believe that functional label-guided molecular discovery may serve as an orthogonal approach to traditional structure-based methods in the pursuit of designing novel functional molecules.

## 2 RELATED WORK

**Labeled chemical datasets.** Chemicals are complex interacting entities, and there are many labels that can be associated with a given chemical. One class is specific protein binding, commonly used to train chemical representation models (Mysinger et al., 2012; Wu et al., 2018). Datasets linking chemicals to their functionality have emerged in recent years (Edwards et al., 2021; Huang et al., 2023; Degtyarenko et al., 2007; Wishart et al., 2023). These datasets were largely compiled from existing databases of well-studied chemicals, limiting their generalizability (Li et al., 2016; Fu et al., 2015). The CheF dataset developed here aims to improve upon these existing datasets by automatically sourcing molecular function from patents to create a high-quality molecular function dataset, ultimately capable of scaling to the entire SureChEMBL database of 32M+ patent-associated molecules (Papadatos et al., 2016). To our knowledge, the full scale-up would create not just the largest chemical function dataset, but rather the largest labeled chemical dataset of any kind. Its high coverage of chemical space means that the CheF dataset, in its current and future iterations, may serve as a benchmark for the global evaluation of chemical representation models.

**Patent-based molecular data mining and prediction.** Building chemical datasets often involves extracting chemical identities, reaction schemes, quantitative drug properties, and chemical-disease relationships (Senger et al., 2015; Papadatos et al., 2016; He et al., 2021; Sun et al., 2021; Magariños et al., 2023; Zhai et al., 2021; Li et al., 2016). We recently used an LLM to extract patent-derived information to help evaluate functional relevance of results from a machine learning-based chemical similarity search (Anonymous et al., 2023). We expand upon previous works through the large-scale LLM-based extraction of broad chemical functionality from a corpus of patent literature. This is a task that LLMs were not explicitly trained to do, and we provide validation results for this approach.

Recent work also focused on molecular generation from chemical subspaces derived from patents containing specific functional keywords, for example, all molecules relating to tyrosine kinase inhibitor activity (Subramanian et al., 2023). This allows for a model that can generate potential tyrosine kinase inhibitors but would need to be retrained to predict molecules of a different functional label. In our work, we focus on label classification rather than molecular generation. Further, we integrate multiple functional labels for any given molecule, allowing us to broadly infer molecular

functionality given structure. Generative models could be trained on the described dataset, allowing for label-guided molecular generation without re-training for each label.

**Chemical-to-textual translation.** Recent work investigated the translation of molecules to descriptive definitions and vice versa (Edwards et al., 2021; 2022; Su et al., 2022). The translation between language and chemical representations is promising as it utilizes chemical relationships implicit in text descriptions. However, decoder-based molecule-text translation models appear to us unlikely to be utilized for novel drug discovery tasks as experimentalists desire strongly deterministic results, reported prediction confidences, and alternative prediction hypotheses. To satisfy these constraints, we opted for a discriminative structure-to-function model.

Many existing chemical-to-text translation models have been trained on datasets containing structural nomenclature and irrelevant words mixed with desirable functional information (Edwards et al., 2021; Degtyarenko et al., 2007). Inclusion of structural nomenclature causes inflated prediction metrics for functional annotation or molecular generation tasks, as structure-to-name and name-to-structure is simpler than structure-to-function and function-to-structure. The irrelevant words may cause artifacts during the decoding process depending on the prompt, skewing results in ways irrelevant to the task. In our work, we ensured our model utilized only chemical structure, and not structural nomenclature, when predicting molecular function to avoid data leakage.

## 3    RESULTS

Patents are an abundant source of highly specific chemical knowledge. It is plausible that a large dataset of patent-derived molecular function would capture most known functional relationships and could approximate the chemical function landscape. High-fidelity approximation of the chemical function landscape would implicitly capture complex physical and biological interactions given that chemical function arises from both a molecule's structure and its interacting partners. This would allow for the prediction of functional labels for chemicals which is, to our knowledge, a novel task.

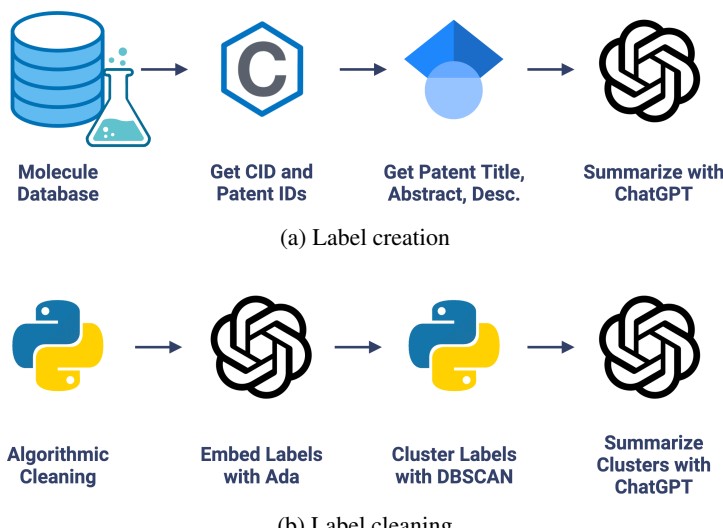

(a) Label creation

(b) Label cleaning

Figure 1: **Chemical function dataset creation.** (a) LLM extracts molecular functional information present in patents into brief labels. Example shown in Figure S2. (b) Chemical functional labels were cleaned with algorithmic-, embedding-, and LLM-based methods.

**Chemical function dataset creation**. We set out to create a large-scale database of chemicals and their patent-derived molecular functionality. To do so, a random 100K molecules and their associated patents were chosen from the SureChEMBL database to create a Chemical Function (CheF) dataset (Fig. S1) (Papadatos et al., 2016). To ensure that patents were highly relevant to their respective molecule, only molecules with fewer than 10 patents were included in the random selection, reducing the number of available molecules by 12%. This was done to exclude over-patented molecules like penicillin with over 40,000 patents, most of which are irrelevant to its functionality.

For each molecule-associated patent in the CheF dataset, the patent title, abstract, and description were scraped from Google Scholar and cleaned. ChatGPT (gpt-3.5-turbo) was used to generate 1–3 functional labels describing the patented molecule given its unstructured patent data (Fig. 1a). The LLM-assisted function extraction method's success was validated manually across 1,738 labels generated from a random 200 CheF molecules. Of these labels, 99.6% had correct syntax and 99.8% were relevant to their respective patent (Table S1). 77.9% of the labels directly described the labeled molecule's function. However, this increased to 98.2% when considering the function of the primary patented molecule, of which the labeled molecule is an intermediate (Table S1).

The LLM-assisted method resulted in 104,607 functional labels for the 100K molecules. These were too many labels to yield any predictive power, so measures were taken to consolidate these labels into a concise vocabulary. The labels were cleaned, reducing the number of labels to 39,854, and further consolidated by embedding each label with a language model (OpenAI's textembedding-ada-002) to group grammatically dissimilar yet semantically similar labels together. The embeddings were clustered with DBSCAN using a cutoff that minimized the number of clusters without cluster quality deterioration (e.g., avoiding the grouping of antiviral, antibacterial, and antifungal) (Fig. S4). Each cluster was summarized with ChatGPT to obtain a single representative cluster label.

The embedding-based clustering and summarization process was validated across the 500 largest clusters. Of these, 99.2% contained semantically common elements and 97.6% of the cluster summarizations were accurate and representative of their constituent labels (Table S2). These labels were mapped back to the CheF dataset, resulting in 19,616 labels (Fig. 1b). To ensure adequate predictive power, labels appearing in less than 50 molecules were dropped. The final CheF dataset consisted of 99,454 molecules and their 1,543 descriptive functional labels (Fig. 1, Table S3).

**Functional labels map to natural clusters in chemical structure space**. Molecular function nominally arises directly from structure, and thus any successful dataset of functional labels should cluster in structural space. This hypothesis was based in part on the observation that chemical function is often retained despite minor structural modifications (Maggiora et al., 2014; Patterson et al., 1996). And due to molecules and their derivatives frequently being patented together, structurally similar molecules should be annotated with similar patent-derived functions. This rationale generally holds, but exceptions include stereoisomers with different functions (e.g. as for thalidomide) and distinct structures sharing the same function (e.g. as for beta-lactam antibiotics and tetracyclines).

To evaluate this hypothesis, we embedded the CheF dataset in structure space by converting the molecules to molecular fingerprints (binary vectors representing a molecule's substructures), visualized with t-distributed Stochastic Neighbor Embedding (t-SNE) (Fig. 2). Then, to determine if the CheF functional labels clustered in this structural space, the maximum fingerprint Tanimoto similarity was computed between the fingerprint vectors of each molecule containing a given label; this approach provides a measure of structural similarity between molecules that have the same functional label (Fig. 2) (Bajusz et al., 2015). This value was compared to the maximum similarity computed from a random equal-sized set of molecules to determine significance. Remarkably, 1,192 of the 1,543 labels were found to cluster significantly in structural space (independent t-tests per label, false-discovery rate of 5%). To give an idea of the meaning of this correlation, inherent clustering was visualized for the labels 'hcv' (hepatitis C virus), 'electroluminescence', 'serotonin', and '5-ht' (5-hydroxytryptamine, the chemical name for serotonin) (Fig. 2). For the label 'electroluminescence' there was one large cluster containing almost only highly conjugated molecules (Fig. 2c). For 'hcv', there were multiple distinct communities representing antivirals targeting different mechanisms of HCV replication. Clusters were observed for NS5A inhibitors, NS3 macrocyclic and peptidomimetic protease inhibitors, and nucleoside NS5B polymerase inhibitors (Fig. 2a, S5). The observed clustering of functional labels in structure space provided evidence that the CheF dataset labels had accurately captured structure-function relationships, validating our initial hypothesis.

**Label co-occurrences reveal the text-based chemical function landscape**. Patents contain joint contextual information on the application, structure, and mechanism of a given compound. We attempted to determine the extent to which the CheF dataset implicitly captured this joint semantic context by assessing the graph of co-occurring functional labels (Fig. 3). Each node in the graph represents a CheF functional label, and their relative positioning indicates the frequency of co-occurrence between labels, with labels that co-occur more frequently placed closer together. To prevent the visual overrepresentation of extremely common labels (i.e., inhibitor, cancer, kinase), each node's size was scaled based on its connectivity instead of the frequency of co-occurrence.

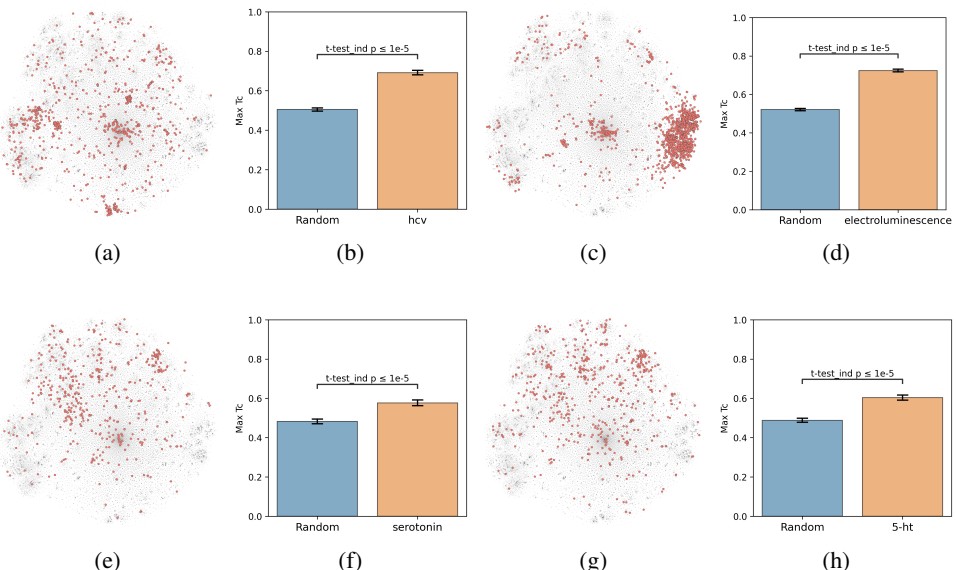

Figure 2: **Text-based functional labels cluster in structural space.** Molecules in the CheF dataset were mapped by their molecular fingerprints and colored based on whether the selected label was present in their set of functional descriptors. The max fingerprint Tanimoto similarity was computed between the fingerprint vectors of each molecule containing a given label and was compared against the max fingerprint Tanimoto similarity from a random equal-sized set of molecules to determine significance to a random control. Many of the labels strongly cluster in structural space, demonstrating that CheF accurately captures structure-function relationships. (a) 'hcv' molecules. (b) 'hcv' degree of clustering. (c) 'electroluminescence' molecules. (d) 'electroluminescence' degree of clustering. (e) 'serotonin' molecules. (f) 'serotonin' degree of clustering. (g) '5-ht' molecules. (h) '5-ht' degree of clustering. See Fig. S5 for more labels.

Modularity-based community detection isolates tightly interconnected groups within a graph, distinguishing them from the rest of the graph. This method was applied to the label co-occurrence graph, with the resulting clusters summarized with GPT-4 into representative labels for unbiased semantic categorization (Table S4, S5, S6). The authors curated the summarized labels for validity and found them representative of the constituent labels; these were then further consolidated for succinct representation of the semantic categorization (Table S4). This revealed a semantic structure in the co-occurrence graph, where distinct communities such as 'Electronic, Photochemical, & Stability' and 'Antiviral & Cancer' could be observed (Fig. 3, Tables S4, S5, S6). Within communities, the fine-grained semantic structure also appeared to be coherent. For example, in the local neighborhood around 'hcv' the labels 'antiviral', 'ns' (nonstructural), 'hbv' (hepatitis B virus), 'hepatitis', 'replication', and 'protease' were found, all of which are known to be semantically relevant to hepatitis C virus (Fig. 3). The graph of patent-derived molecular functions is a visual representation of the text-based chemical function landscape, and represents a potentially valuable resource for linguistic evaluation of chemical function and ultimately drug discovery.

**Coherence of the text-based chemical function landscape in chemical structure space**. To assess how well text-based functional relationships align with structural relationships, the overlap between the molecules of a given label and those of its 10 most commonly co-occurring labels was calculated (Fig. 4). This was achieved by computing the maximum fingerprint Tanimoto similarity from each molecule containing a given label to each molecule containing any of the 10 most commonly co-occurring labels (with <1,000 total abundance). This value was compared to the maximum similarity computed from each molecule containing a given label to a random equal-sized set of molecules to determine significance. This comparison indicated that molecules containing the 10 most commonly co-occurring labels were closer to the given label's molecules in structure space than a random set for 1,540 of the 1,543 labels (independent t-tests per label, false-discovery rate

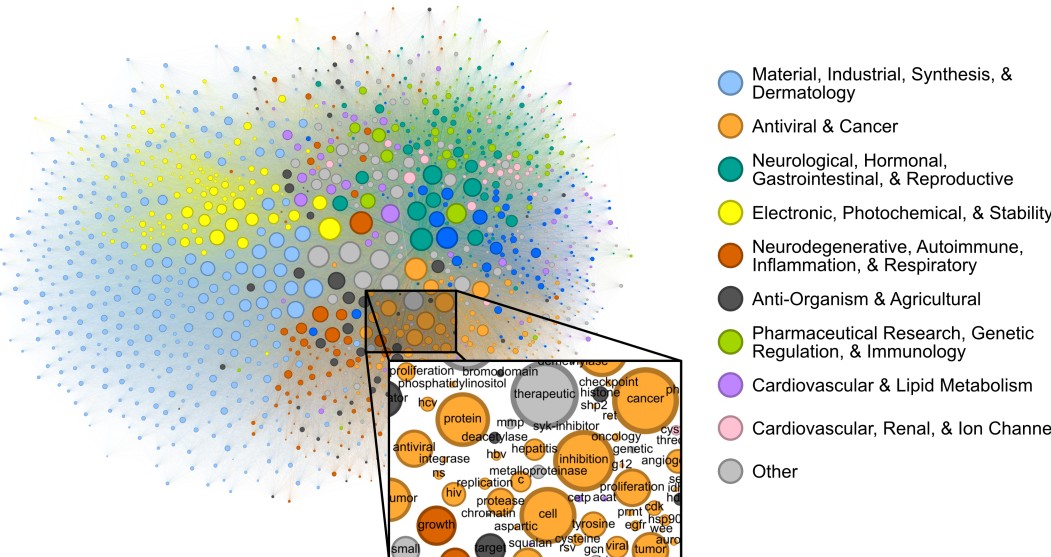

Figure 3: **Label co-occurrences reveal the text-based chemical function landscape.** Node sizes correspond to number of connections, and edge sizes correspond to co-occurrence frequency in the CheF dataset. Modularity-based community detection was used to obtain 19 distinct communities. The communities broadly coincided with the semantic meaning of the contained labels, the largest 10 of which were summarized to representative categorical labels (Tables S4, S5, S6).

of 5%), meaning that text-based functional relationships align with structural relationships (Fig. 4). With the discovery of semantically structured communities, above, this suggests that users can move between labels to identify new compounds and vice versa to assess a compound's function.

**Functional label-guided drug discovery**. To employ the text-based chemical function landscape for drug discovery, multi-label classification models were trained on CheF to predict functional labels from molecular fingerprints (Table S7). The best performing model was a logistic regression model on molecular fingerprints with positive predictive power for 1,532/1,543 labels and >0.90 ROC-AUC for 458/1,543 labels (Fig. 5a).

This model can thus be used to comprehensively annotate chemical function, even when existing annotations are fragmented or incomplete. As an example, for a known hepatitis C antiviral the model strongly predicted 'antiviral', 'hcv', 'ns' (nonstructural) (94%, 93%, 70% respectively) while predicting 'protease' and 'polymerase' with low confidence (0.02%, 0.00% respectively) (Fig. 5b). The low-confidence 'protease' and 'polymerase' predictions suggested that the likely target of this drug was the nonstructural NS5A protein, rather than the NS2/3 proteases or NS5B polymerase, a hypothesis that has been validated outside of patents in the scientific literature (Ascher et al., 2014).

The ability to comprehensively predict functional profiles allows for the discovery of new drugs. For example, the label 'serotonin' was used to query the test set predictions, and a ranked list of the 10 molecules most highly predicted for 'serotonin' were obtained (Fig. 5c). All ten of these were patented in relation to serotonin: 8 were serotonin receptor ligands (5-HT1, 5-HT2, 5-HT6) and 2 were serotonin reuptake inhibitors. Similarly, the synonymous label '5-ht' was used as the query and the top 10 molecules were again obtained (Fig. 5d). Of these, seven were patented in relation to serotonin (5-HT1, 5-HT2, 5-HT6), four of which were also found in the aforementioned 'serotonin' search. The remaining three molecules were patented without reference to the serotonin receptor, but were instead patented for depressant, anti-anxiety, and memory dysfunction relieving effects, all of which have associations with serotonin and its receptor. The identification of known serotonin receptor ligands, together with the overlapping results across synonymous labels, provides an internal validation of the model. Additionally, these search results suggest experiments in which the "mispredicted" molecules may bind to serotonin receptors or otherwise be synergistic with the function of serotonin, thereby demonstrating the practical utility of moving with facility between chemicals and their functions.

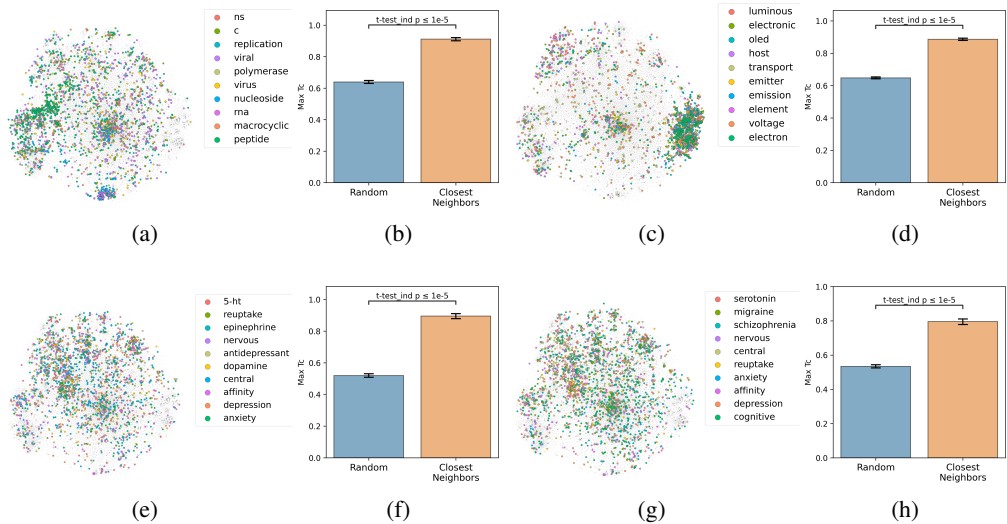

Figure 4: **Coherence of the text-based chemical function landscape in structure space.** To assess the alignment of text-based functional relationships with structural relationships, the max fingerprint Tanimoto similarity from each molecule containing a given label to each molecule containing any of its 10 most frequently co-occurring labels ($<$1,000 total abundance) was compared against the max fingerprint Tanimoto similarity to a random subset of molecules of the same size. (a) 'hcv' neighboring labels' molecules. (b) Degree of coincidence between 'hcv' and neighboring labels. (c) 'electroluminescence' neighboring labels' molecules. (d) Degree of coincidence between 'electroluminescence' and neighboring labels. (e) 'serotonin' neighboring labels' molecules. (f) Degree of coincidence between 'serotonin' and neighboring labels. (g) '5-ht' neighboring labels' molecules. (h) Degree of coincidence between '5-ht' and neighboring labels. See Fig. S5 for more labels.

To examine the best model's capability in drug repurposing, functional labels were predicted for 3,242 Stage-4 FDA approved drugs (Fig. S7) (Ochoa et al., 2021). Of the 16 drugs most highly predicted for 'hcv', 15 were approved Hepatitis C Virus (HCV) antivirals. Many of the mispredictions in the top 50 were directly relevant to HCV treatment including 8 antivirals and 8 polymerase inhibitors. The remaining mispredictions included 3 ACE inhibitors and 2 BTK inhibitors, both of which are peripherally associated with HCV through liver fibrosis mitigation and HCV reactivation, respectively (Corey et al., 2009; Mustafayev & Torres, 2022). Beyond showing its power, this example suggests that functional label-guided drug discovery may serve as a useful paradigm for rapid antiviral repurposing to mitigate future pandemics.

## 4 DISCUSSION

While *in silico* drug discovery often proceeds through structural and empirical methods such as protein-ligand docking, receptor binding affinity prediction, and pharmacophore design, we set out to investigate the practicality of orthogonal methods that leverage the extensive corpus of chemical literature. To do so, we developed an LLM- and embedding-based method to create a Chemical Function (CheF) dataset of 100K molecules and their 631K patent-derived functional labels. Over 78% of the functional labels corresponded to distinct clusters in chemical structure space, indicating congruence between chemical structures and individual text-derived functional labels. Moreover, there was a semantically coherent text-based chemical function landscape intrinsic to the dataset that was found to correspond with broad fields of functionality. Finally, it was found that the relationships in the text-based chemical function landscape mapped with high fidelity to chemical structure space (99.8% of labels), indicating approximation to the actual chemical function landscape.

To leverage the chemical function landscape for drug discovery, several models were trained and benchmarked on the CheF dataset to predict functional labels from molecular fingerprints (Table. S7). The top-performing model was utilized for practical applications such as unveiling an undis-

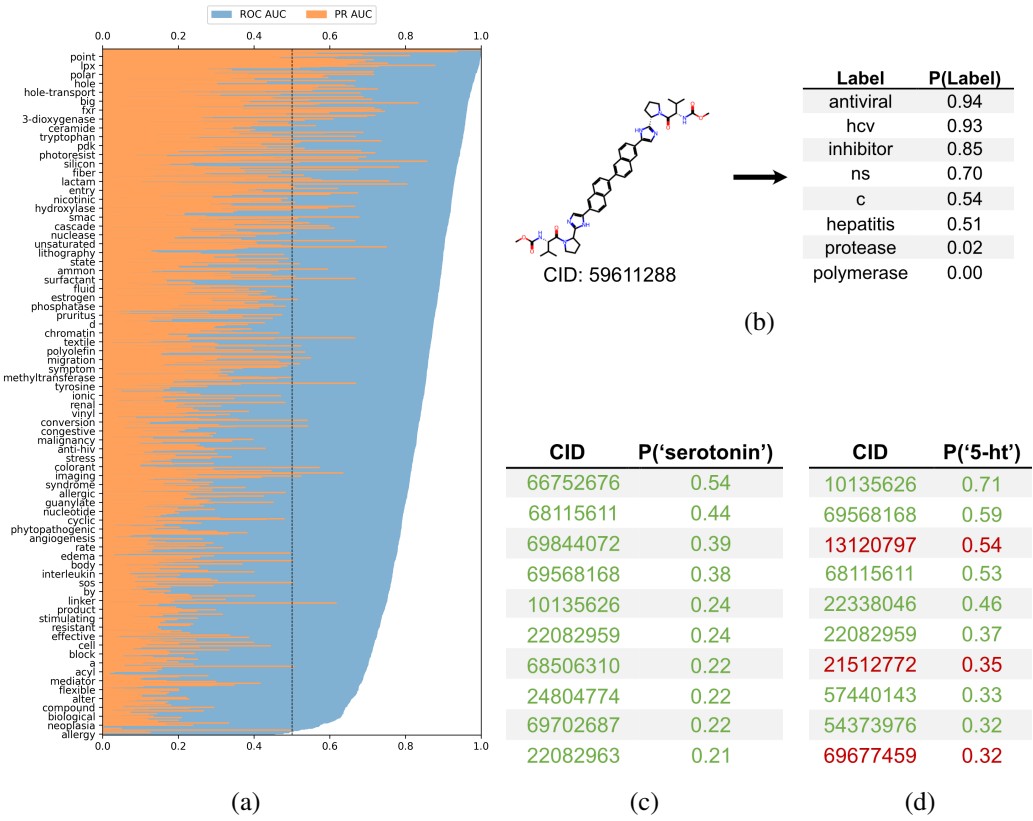

(a)                    (c)                    (d)

Figure 5: **Functional label-guided drug discovery.** (a) Test set results from best-performing model that predicts functional labels from molecular fingerprints. Labels sorted by ROC-AUC, showing every 20 labels for clarity. Black line indicates ROC-AUC random threshold. Average test ROC-AUC and PR-AUC were 0.84 and 0.20, respectively. (b) Model-based comprehensive annotation of chemical function. Shown is a test set molecule patented for hepatitis C antiviral treatment. The highly predicted 'hcv', 'ns', and 'inhibitor' with the low-predicted 'protease' and 'polymerase' can be used to infer that the drug acts on NS5A to inhibit HCV replication, revealing a mechanism undisclosed in the patent. (c-d) Functional label-based drug candidate identification, showcasing the top 10 test set molecules for 'serotonin' or '5-ht'; true positives in green, false positives in red. The false positives offer potential for drug discovery and repurposing, especially when considering these have patents for related neurological uses (i.e., anti-anxiety and memory dysfunction).

closed drug mechanism, identifying novel drug candidates, and mining FDA-approved drugs for repurposing and combination therapy uses. Since the CheF dataset is scalable to the entire 32M+ molecule database, we anticipate that many of these predictions will only get better into the future.

The CheF dataset inherently exhibits a bias towards patented molecules. This implies sparse representation of chemicals with high utility but low patentability, and allows for false functional relationships to arise from prophetic claims. Additionally, by restricting the dataset to chemicals with <10 patents, it neglects important well-studied molecules like Penicillin. The inclusion of over-patented chemicals could be accomplished by using only the most abundant k terms for a given molecule, using a fine-tuned LLM to only summarize patents relevant to molecular function (ignoring irrelevant patents on applications like medical devices), or employing other data sources like PubChem or PubMed to fill in these gaps. Increasing label quality and ignoring extraneous claims might be achieved through an LLM fine-tuned on high-quality examples. Further quality increases may result from integration of well-documented chemical-gene and chemical-disease relationships into CheF.

The analysis herein suggests that a sufficiently large chemical function dataset contains a text-based function landscape that approximates the actual chemical function landscape. Further, we demonstrate one of the first examples of functional label-guided drug discovery, made possible utilizing

state-of-the-art advances in machine learning. Models in this paradigm have the potential to automatically annotate chemical function, examine non-obvious features of drugs such as side effects, and down-select candidates for high-throughput screening. Moving between textual and physical spaces represents a promising paradigm for drug discovery in the age of machine learning.

## 5  METHODS

**Database creation.**  The SureChEMBL database was shuffled and converted to chiral RDKit-canonicalized SMILES strings, removing malformed strings (Weininger, 1988; Papadatos et al., 2016; Landrum et al., 2013). SMILES strings were converted to InChI keys and used to obtain PubChem CIDs (Kim et al., 2023). To minimize costs and prevent label dilution, only molecules with fewer than 10 patents were included. This reduced the dataset from 32M to 28.2M molecules, a 12% decrease. A random 100K molecules were selected as the dataset. For each associated patent, the title, abstract, and description were scraped from Google Scholar and cleaned.

The patent title, abstract, and first 3500 characters of the description were summarized into brief functional labels using ChatGPT (gpt-3.5-turbo) from July 15th, 2023, chosen for low cost and high speed. Cost per molecule was $0.005 using gpt-3.5-turbo. Responses from ChatGPT were converted into sets of labels and linked to their associated molecules. Summarizations were cleaned, split into individual words, converted to lowercase, and converted to singular if plural. The cleaned dataset resulted in 29,854 unique labels for 99,454 molecules. Fetching patent information and summarizing with ChatGPT, this method's bottleneck, took 6 seconds per molecule with 16 CPUs in parallel. This could be sped up to 3.9 seconds by summarizing per-patent rather than per-molecule to avoid redundant summarizations, and even further to 2.6 seconds by using only US and WO patents.

To consolidate labels by semantic meaning, the vocabulary was embedded with OpenAI's textembedding-ada-002 and clustered to group labels by embedding similarity. DBSCAN clustering was performed on the embeddings with a sweeping epsilon (Ester et al., 1996). The authors chose the epsilon for optimal clustering, set to be at the minimum number of clusters without quality degradation (e.g., avoiding the merging of antiviral, antibacterial, and antifungal). The optimal epsilon was 0.34 for the dataset herein, consolidating down from 29,854 to 20,030 labels. Representative labels for each cluster were created using gpt-3.5-turbo. The labels from a very large cluster of only IUPAC structural terms were removed to reduce non-generalizable labels. Labels appearing in <50 molecules were dropped to ensure sufficient predictive power. This resulted in a 99,454-molecule dataset with 1,543 unique functional labels, deemed the Chemical Function (CheF) dataset.

**Text-based functional landscape graph.**  Per-molecule label co-occurrence was counted across CheF. Counts were used as edge weights between label nodes to create a graph, visualized in Gephi using force atlas, nooverlap, and label adjust methods (default parameters) (Bastian et al., 2009). Modularity-based community detection with 0.5 resolution resulted in 19 communities.

**Coincidence of labels and their neighbors in structure space.** The 100K molecular fingerprints were t-SNE projected using sckit-learn, setting the perplexity parameter to 500. Molecules were colored if they contained a given label, see chefdb.app. The max fingerprint Tanimoto similarity from each molecule containing a given label to each molecule containing any of the 10 most commonly co-occurring labels was computed. The null co-occurrence was calculated by computing the max similarity from each molecule containing a given label to a random equal-sized set. Significance for each label was computed with an independent 2-sided t-test. The computed P values were then subjected to a false-discovery-rate (FDR) correction and the labels with P < 0.05 after FDR correction were considered significantly clustered (Benjamini & Hochberg, 1995). Limiting max co-occurring label abundance to 1K molecules was necessary to avoid polluting the analysis, as hyper-abundant labels would force the Tanimoto similarity to 1.0.

**Model training.**  Several multi-label classification models were trained to predict the CheF from molecular representations. These models included logistic regression (C=0.001, max_iter=1000), random forest classifier (n_estimators=100, max_depth=10), and a feedforward neural network (BCEWithLogitsLoss, layer sizes (512, 256), 5 epochs, 0.2 dropout, batch size 32, learning rate 0.001; 5-fold CV to determine params). A random 10% test set was held out from all model training. Macro average and individual label ROC-AUC and PR-AUC were calculated.

ETHICS STATEMENT

Consideration of ML chemistry dual use often focuses on the identification of toxic chemicals and drugs of abuse. As patents typically describe the beneficial applications of molecules, it is unlikely that a model trained on CheF labels will be able to identify novel toxic compounds. Functional labels for the chemical weapons VX and mustard gas were predicted from our model, found to contain no obvious indications of malicious properties. On the contrary, drugs of abuse were more easily identifiable, as the development of neurological compounds remains a lucrative objective. 5-MeO-DMT, LSD, fentanyl, and morphine all had functional labels of their primary mechanism predicted with moderate confidence. However, benign molecules also predicted these same labels, indicating that it may be quite challenging to intentionally discover novel drugs of abuse using the methods contained herein.

REPRODUCIBILITY STATEMENT

The CheF dataset has been made publicly available under the MIT license at https://doi.org/10.5281/zenodo.8350193. An interactive visualization of the dataset can be found at chefdb.app.

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

## A  PROMPTS

**Patent summarization.** The system prompt used was "You are an organic chemist summarizing chemical patents", and the user prompt was "Return a short set of three 1-3 word descriptors that best describe the chemical or pharmacological function(s) of the molecule described by the given patent title, abstract, and partial description (giving more weight to title & abstract). Be specific and concise, but not necessarily comprehensive (choose a small number of great descriptor). Follow the syntax '{descriptor_1} / {descriptor_2} / {etc}', writing 'NA' if nothing is provided. DO NOT BREAK THIS SYNTAX. The following is the patent:", followed by the patent title, abstract, and partial description.

**Word embedding cluster summarization.** Each cluster's labels were fed into GPT-3.5-turbo with the system prompt "You are a PhD pharmaceutical chemist" and the user prompt: "Given a set of molecular descriptors, return a single descriptor representing the centroid of the terms. Do not speculate. Only use the information provided. Be concise, not explaining answers. Example 1 Set of Descriptors: 11(beta)-hsd1, 11-hsd-2, 17$\beta$-hsd3 Example 1 Average Descriptor: hsd Example 2 Set of Descriptors: anti-retroviral, anti-retrovirus, anti-viral, anti-virus, antiretroviral, antiretrovirus, antiviral, antivirus Example 2 Average Descriptor: antiviral Set of Descriptors: ＿INSERT_DESCRIPTORS_HERE＿ Average Descriptor:".

**Graph label cluster summarization.** Each cluster's labels were fed into GPT-4 with the system prompt "You are a PhD pharmaceutical chemist" and the user prompt: "Pretend you are a pharmaceutical chemist. I will provide you with several terms, and your job is to summarize the terms into appropriate categories. Be succinct, focusing on the broadest categories while still being representative. Don't show your work. Example terms: Antiviral HCV Kinase Cancer Polymerase Protease Example summarization: Antiviral & Cancer Terms: ＿INSERT_DESCRIPTORS_HERE＿ Summarization:"".

## B  SUPPLEMENTAL DATA

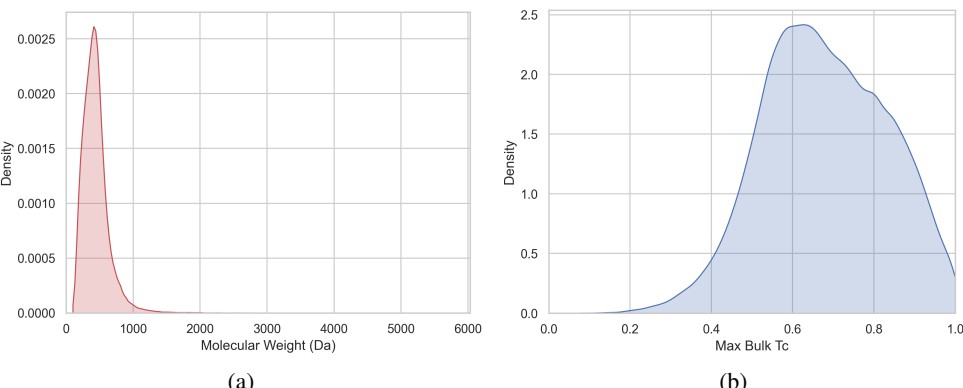

(a)

(b)

Figure S1: **Molecular weight and structural similarity distribution of the CheF dataset.** (a) Molecular weight of each molecule in the dataset. Minimum: 100.12 Da; Maximum: 5749.60 Da; Mean: 440.79 Da; Std: 203.96 Da. (b) Maximum bulk fingerprint Tanimoto coefficient (Tc) for each molecule in the dataset. Bulk Tc measures how similar a given molecule's structure is to all of the other molecules in the dataset. Max Bulk Tc returns the structural similarity of a molecule to the most structurally similar molecule in the dataset. High Max Bulk Tc indicates redundant structures, mid-low Max Bulk Tc indicates diverse structures. Minimum: 0.076; Maximum: 1.00; Mean: 0.68; Std: 0.15.

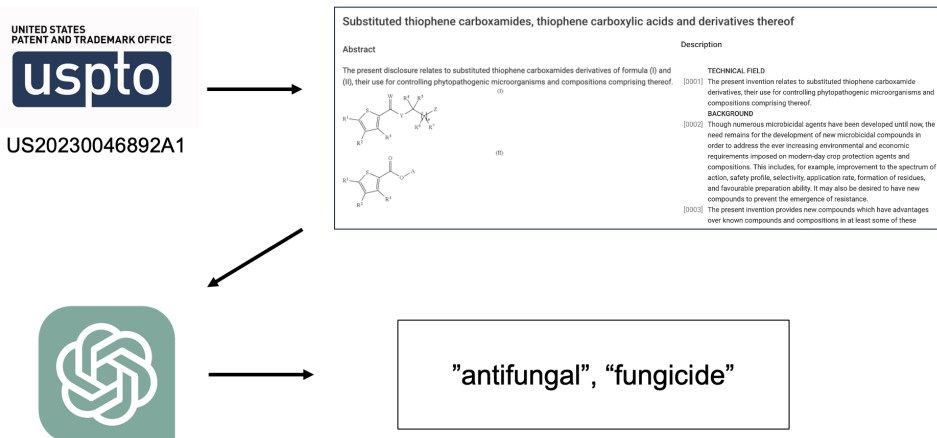

Figure S2: **Example of LLM-based chemical function extraction.** Patent IDs are used to retrieve the patent title, abstract, and description from Google Scholar. ChatGPT is then prompted to extract out the chemical function of the molecule being described by the patent.

Table S1: **ChatGPT patent summarization validation.** Manual validation was performed on 200 molecules randomly chosen from the CheF dataset. These 200 molecules had 596 valid associated patents, and 1,738 ChatGPT summarized labels. These labels were manually validated to determine the ratio of correct syntax, relevance to patent, and relevance to the Molecule of Interest (MOI).

| Validation Task | Fraction Correct |
|---|---|
| Syntax | 0.996 |
| Label relevant to patent | 0.998 |
| Label refers to MOI, target of MOI, or downstream effects of MOI | 0.779 |
| Label refers to MOI, target of MOI, downstream effects of MOI, or molecules of which MOI is an intermediate | 0.982 |

Table S2: **Validation of ChatGPT-aided label consolidation.** The first 500 of the 3,178 clusters of greater than one label (sorted in descending cluster size order) were evaluated for whether or not the clusters contained semantically common elements. The ChatGPT consolidated cluster labels were then analyzed for accuracy and representativeness. Common failure modes for clustering primarily included the grouping of grammatically similar, but not semantically similar labels (e.g., ahas-inhibiting, ikk-inhibiting). Failure modes for ChatGPT commonly included averaging the terms to the wrong shared common element (e.g., anti-fungal and anti-mycotic being consolidated to the label "anti").

| Validation Task | Fraction Correct |
|---|---|
| Cluster contains semantically common elements | 0.992 |
| ChatGPT cluster summarization accurate & representative | 0.976 |

Table S3: **Comparison of Chemical-Text Datasets.** Comparison of CheF to existing chemical-text datasets ChEBI and ChemFOnt (Degtyarenko et al., 2007; Wishart et al., 2023) by current size (# molecules), maximum automated scaleup size (# molecules), text-type, whether or not structure and function are separate in the text, and the data source used for dataset construction. Both ChEBI and ChemFOnt were built from existing datasets with additional manual curation and annotation, limiting potential automated scaleup size. In contrast, the method used to build CheF scales readily, allowing for a potential dataset size of 32M molecules.

| Dataset | Curr. Size | Scaleup Size | Text-Type | S/F Separate | Data Source |
|---|---|---|---|---|---|
| ChEBI | 103K | 103K+ | Long text | No | DB Agg. / Manual |
| ChemFOnt | 342K | 1M+ | Labels | Yes | DB Agg. / Manual |
| CheF (ours) | 100K | 32M+ | Labels | Yes | LLM-Sum. Patents |

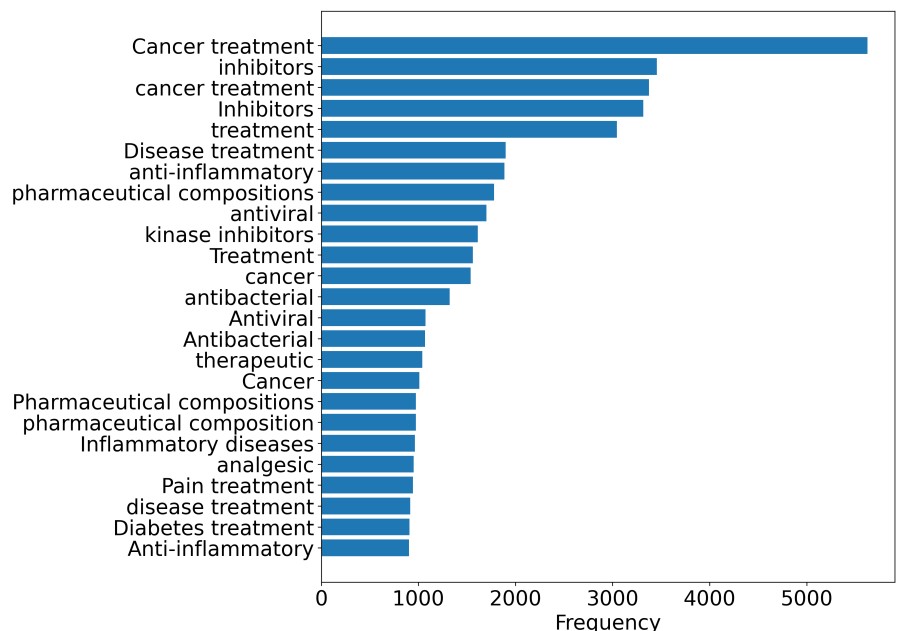

Figure S3: **Most frequent patent summarizations.** The most frequent patent summarizations do not immediately exhibit any dataset-independent biases. The bias towards broad treatment terms, such as cancer, antiviral, and analgesic, likely emerged because these are desirable target functions and are thus overrepresented in patents.

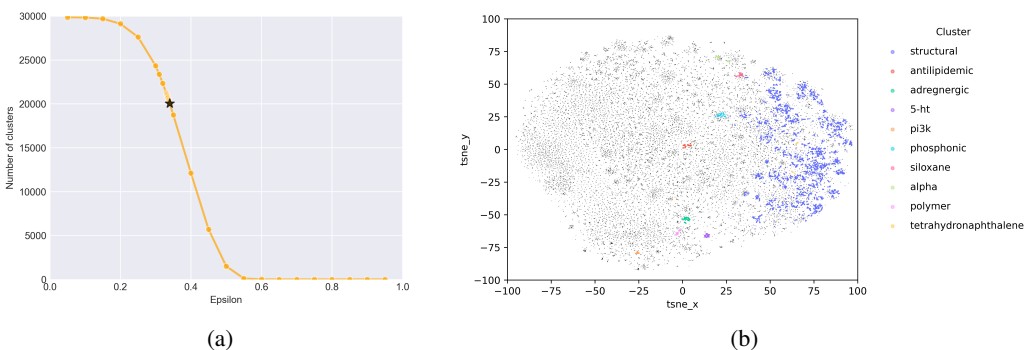

(a)                                              (b)

Figure S4: **DBSCAN clustering on Ada-002 text embeddings reduces the number of labels.** (a) The optimal DBSCAN epsilon value was defined as the cutoff resulting in the smallest number of clusters without overtly false categories appearing (e.g., merging antiviral, antibacterial, & antifungal). The optimal epsilon was found to be 0.340 for the dataset considered herein (marked by black star), resulting in a consolidation from 29,854 labels to 20,030 clusters. The labels in each cluster were then consolidated with ChatGPT, creating a set of 20,030 labels. (b) t-SNE of the Ada-002 text embeddings, colored by the top 10 largest clusters. The largest cluster, found to be all IUPAC structural terms, was removed from the dataset to reduce excessive non-generalizable labels.

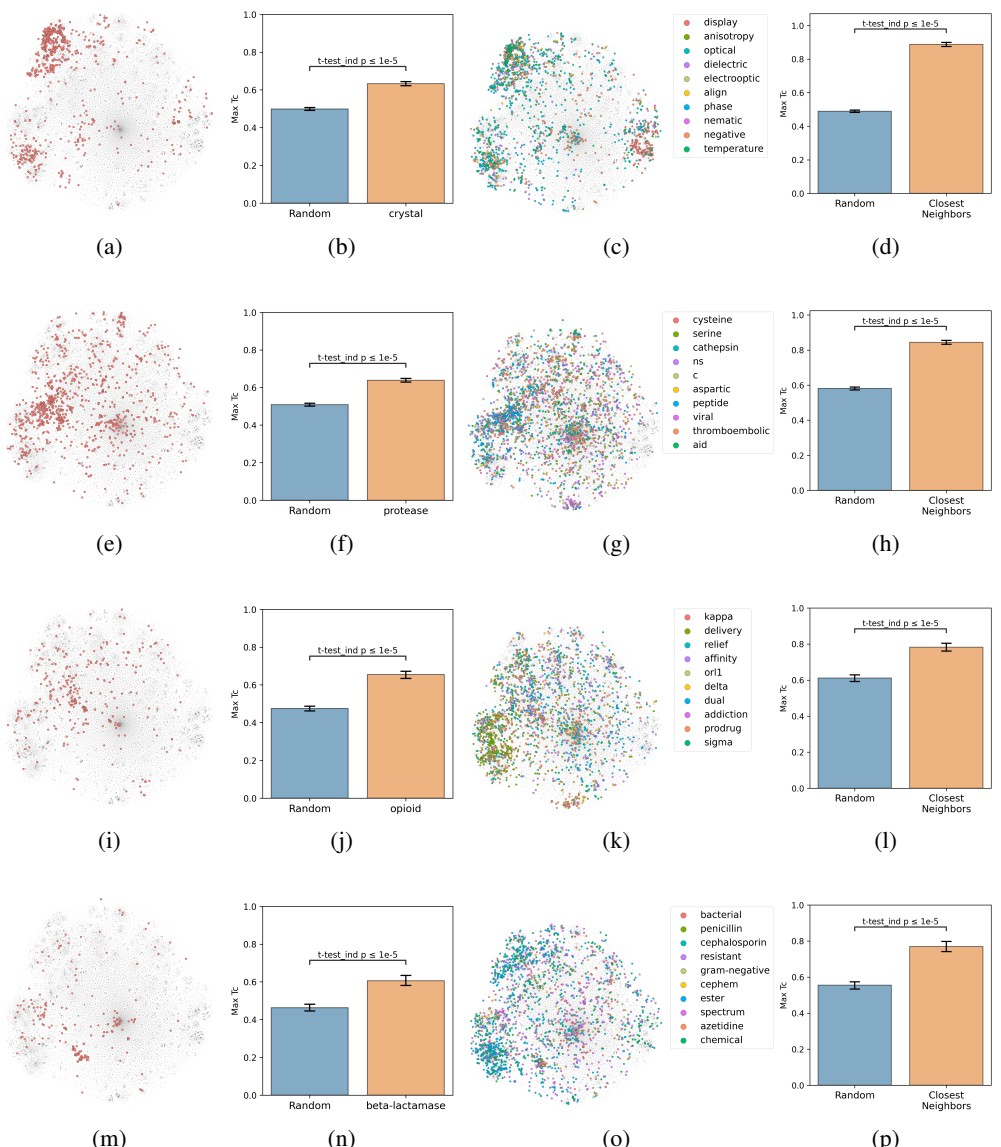

Figure S5: **Additional CheF labels and their clusters in structure space.** Molecules in the CheF dataset were projected based on molecular fingerprints and colored if the selected label was contained by the molecule's set of descriptors. To measure degree of clustering for a single label, the max fingerprint Tanimoto similarity from each molecule containing the selected label, to the other molecules containing that label, compared against the max fingerprint Tanimoto similarity for a random subset of molecules of the same size was obtained, whereas to measure the coincidence between the primary and co-occurring labels, the max fingerprint Tanimoto similarity from each molecule containing the primary label to each molecule containing any of the 10 nearest neighbor labels was compared against the max fingerprint Tanimoto similarity to a random subset of molecules of the same size. (a) Molecules containing label 'crystal'. (b) Degree of clustering for 'crystal'. (c) Molecules containing neighboring labels to 'crystal'. (d) Degree of coincidence between 'crystal' and its neighboring labels. (e) Molecules containing label 'protease'. (f) Degree of clustering for 'protease'. (g) Molecules containing neighboring labels to 'protease'. (h) Degree of coincidence between 'protease' and its neighboring labels. (i) Molecules containing label 'opioid'. (j) Degree of clustering for 'opioid'. (k) Molecules containing neighboring labels to 'opioid'. (l) Degree of coincidence between 'opioid' and its neighboring labels. (m) Molecules containing label 'beta-lactamase'. (n) Degree of clustering for 'beta-lactamase'. (o) Molecules containing neighboring labels to 'beta-lactamase'. (p) Degree of coincidence between 'beta-lactamase' and its neighboring labels.

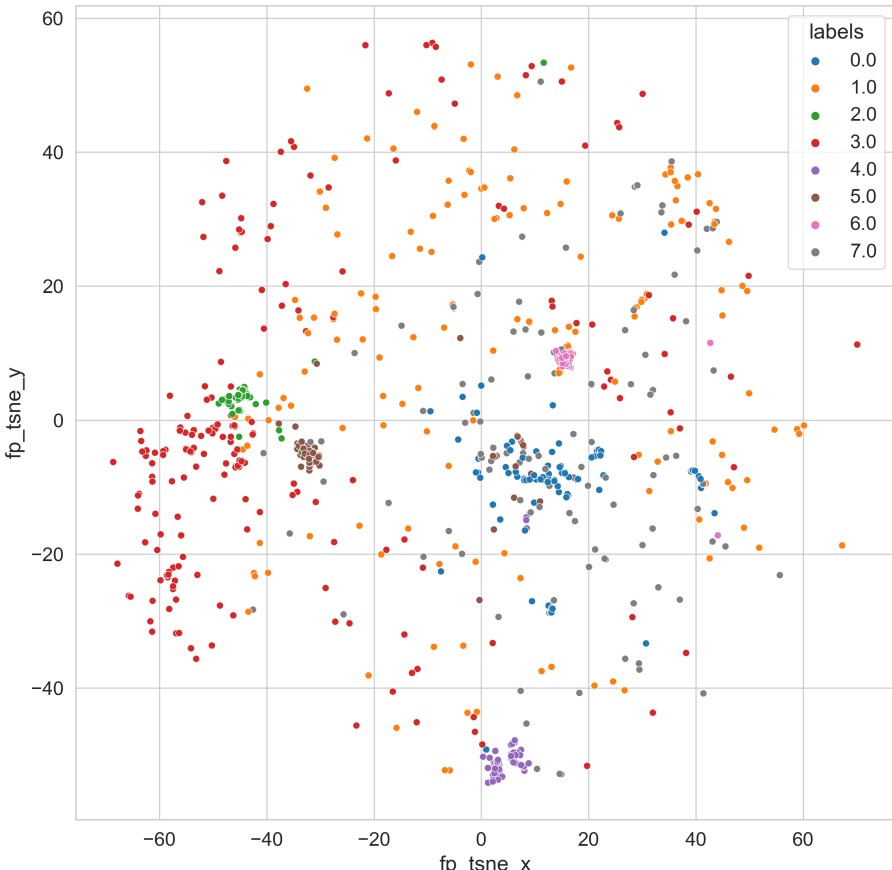

Figure S6: **K-means clustering on molecules containing 'hcv' elucidates Hepatitis C Virus (HCV) antiviral modalities.** The top 20 most frequently occurring labels were obtained for each of 8 clusters to determine their modalities (if applicable). Cluster 4 was the only cluster to contain 'nucleoside' (n=65) and 'nucleotide' (n=12) in the top 20 labels, indicating this cluster primarily contained HCV antiviral nucleoside derivatives likely inhibiting the NS5B polymerase. Cluster 2 contained 'protease' (n=85), 'peptide' (n=35), and 'serine' (n=15), indicating that this cluster primarily contained peptidomimetic protease inhibitors acting on the NS3 serine protease. Cluster 5 contained 'protease' (n=108), 'macrocyclic' (n=42), and serine (n=8), indicating that this cluster contained macrocyclic compounds acting likely as NS3 serine protease inhibitors. Cluster 6 contained no specific mechanistic terms, alluding to the possible mechanism of these molecules inhibiting the NS5A protein.

Table S4: **GPT-4 graph community summarizations.** All labels from the ten most abundant clusters were fed into GPT-4 for categorical summarization. These outputs were verified to be representative of the labels, and were further consolidated by the authors into concise categories.

| GPT4 Cluster summary | Label in graph |
|---|---|
| Chemical Processes & Reactions, Materials & Substances, Photographic & Printing Processes, Cosmetic & Dermatological Applications, Industrial Manufacturing & Production, Sensory Properties | Material, Industrial, Synthesis, & Dermatology |
| Antiviral, Cancer, Cellular Processes, Enzymes, Immunology, Oncology, Protein Interactions, Therapy & Drug Development | Antiviral & Cancer |
| Pain Management, Hormonal Regulation, Gastrointestinal Conditions, Neurological Conditions, Reproductive Health, Obesity Management, Addiction Treatment, Sleep Disorders, Immune Response, Cardiovascular Conditions | Neurological, Hormonal, Gastrointestinal, & Reproductive Health |
| Chemical Compounds & Materials, Electronic & Optoelectronic Devices, Energy & Efficiency, Light & Emission Properties, Stability & Durability, Quantum & Thermodynamics | Electronic, Photochemical, & Stability |
| Neurodegenerative Diseases, Inflammatory & Autoimmune Diseases, Respiratory Diseases, Immune Response & Regulation, Enzymes & Mediators, Drug Development & Therapeutics | Neurodegenerative, Autoimmune, Inflammation, & Respiratory |
| Antibacterial, Antifungal, Antiparasitic, Antimalarial, Antimicrobial, Antiprotozoal, Antitubercular, Insecticide, Herbicide, Fungicide, Pesticide, Acaricide, Nematicidal, Agricultural & Health Protection | Anti-Organism & Agricultural |
| Drug Development & Delivery, Diagnostic & Monitoring, Gene & Protein Regulation, Epigenetics & Transcription, Immunology & Vaccines | Pharmaceutical Research, Genetic Regulation, Immunology |
| Neurological & Psychiatric Disorders, Cognitive & Memory Function, Neuropharmacology & Neurotransmission, Mood & Mental Health, Urologic & Sexual Health | Neurological & Urologic |
| Lipid Metabolism & Cardiovascular Health, Diabetes Management, Organ Health & Protection | Cardiovascular & Lipid Metabolism |
| Cardiovascular & Renal Disorders, Ion Channels & Transporters, Anesthetics & Muscle Relaxants, Neurological Disorders & Eye Conditions | Cardiovascular, Renal, & Ion Channel |

Table S5: **Arbitrary 20 CheF labels from each summarized co-occurrence neighborhood.** Modularity-based community detection was performed on the CheF co-occurrence graph to obtain 19 distinct communities. The communities appeared to broadly coincide with the semantic meaning of the contained labels, and the largest 10 communities were summarized to a common label. Shown are a random 20 labels from the first five summarized communities.

| Material, Industrial, Synthesis, & Dermatology | Antiviral & Cancer | Neurological, Hormonal, Gastrointestinal, & Reproductive Health | Electronic, Photochemical, & Stability | Neuro-degenerative, Autoimmune, Inflammation, & Respiratory |
|---|---|---|---|---|
| absorb | aid | analgesic | carbazole | activate |
| acid | antiviral | condition | compound | adhesion |
| binder | c | ligand | expand | alzheimer |
| care | cancer | modulate | life | amyloid |
| cosmetic | cell | modulator | light | anti-inflammatory |
| destabilize | g12 | p2x7 | material | autoimmune |
| form | hbv | pain | activated | cox |
| functional | hcv | prophylaxis | amine | disease |
| ionic | hepatitis | prostate | anisotropy | elastase |
| method | hiv | receptor | aromatic | il-17 |
| modification | inhibit | relief | blue | inflammation |
| optical | inhibition | selective | capability | inflammatory |
| photochromic | inhibitor | tgr5 | characteristic | interferon |
| plastic | integrase | tract | charge | lung |
| polymer | kinase | treatment | condensed | neuro-degenerative |
| preserve | kras | various | crystal | neuro-inflammation |
| production | mapk | 5-ht | cyclic | sting |
| protective | nucleoside | 7 | device | airway |
| sensitivity | phosphatidyl-inositol | addiction | dielectric | allergic |
| skin | phosphorylation | adrenergic | diode | allergy |

Table S6: **Arbitrary 20 CheF labels from each summarized co-occurrence neighborhood.** Modularity-based community detection was performed on the CheF co-occurrence graph to obtain 19 distinct communities. The communities appeared to broadly coincide with the semantic meaning of the contained labels, and the largest 10 communities were summarized to a common label. Shown are a random 20 labels from the second five summarized communities.

| Anti-Organism & Agricultural | Pharmaceutical Research, Genetic Regulation, Immunology | Neurological & Urologic | Cardiovascular & Lipid Metabolism | Cardiovascular, Renal, & Ion Channel |
|---|---|---|---|---|
| amide | assay | anticonvulsant | carbonic | cardiovascular |
| control | bind | cerebral | ischemia | channel |
| derivative | bromodomain | disorder | level | ion |
| infection | diagnostic | function | liver | stroke |
| protection | drug | mitochondrial | prevention | ace |
| acaricide | potential | neural | reducer | anesthetic |
| acetic | psma | neuroprotective | reducing | angina |
| animal | regulator | pde | reduction | angiotensin |
| anti | sirtuin | schizophrenia | regulate | anti-hypertensive |
| anti-malarial | targeting | sedative | releasing | blocker |
| anti-microbial | 6 | system | retinoid | calcium |
| antiparasitic | alter | urologic | vap | cardiac |
| aryl | analog | 4 | vascular | cardiotonic |
| azetidin | atp | 5 | a | circulation |
| azetidine | atrophy | anti-psychotic | aldose | c-transport |
| bacterial | bioavailability | antidepressant | alleviate | contraction |
| bactericide | biological | antitussive | antilipidemic | diuretic |
| beta-lactamase | biomarker | anxiolytic | blood | failure |
| bicyclic | combinatorial | brain | cholesterol | heart |
| bridge | cytotoxic | central | cholesterolemia | hypertensive |

Table S7: **Fingerprint models benchmarked on CheF.** To assess a baseline benchmark on the CheF dataset of ∼100K molecules, several molecular fingerprint-based models were trained on 90% of the training data and evaluated on the 10% test set holdout. Macro average ROC-AUC and PR-AUC was calculated across all 1,543 labels. Logistic regression (LR), random forest classifier (RFC), and a 2-layer feedforward neural network (FFN) were trained. Parameters for LR and RFC were chosen to be common default values, whereas the FFN layer number and size were chosen through a 5-fold cross validation.

| Model | ROC-AUC | PR-AUC |
|---|---|---|
| FP + LR | **0.84** | **0.20** |
| FP + RFC | 0.80 | 0.13 |
| FP + FFN | 0.81 | 0.12 |

| Name | P(hcv) | P(hepatitis) | P(antiviral) | P(ns) | P(protease) | P(polymerase) | P(ace) | P(btk) |
|---|---|---|---|---|---|---|---|---|
| DACLATASVIR DIHYDROCHLORIDE | 0.95 | 0.42 | 0.95 | 0.84 | 0.02 | 0.00 | 0.00 | 0.01 |
| DACLATASVIR | 0.95 | 0.42 | 0.95 | 0.84 | 0.02 | 0.00 | 0.00 | 0.01 |
| GRAZOPREVIR | 0.94 | 0.41 | 0.81 | 0.59 | 0.73 | 0.00 | 0.00 | 0.01 |
| BOCEPREVIR | 0.91 | 0.73 | 0.42 | 0.03 | 0.88 | 0.00 | 0.00 | 0.00 |
| PARITAPREVIR | 0.83 | 0.12 | 0.75 | 0.67 | 0.73 | 0.00 | 0.00 | 0.00 |
| SIMEPREVIR | 0.83 | 0.06 | 0.35 | 0.09 | 0.07 | 0.01 | 0.00 | 0.00 |
| SIMEPREVIR SODIUM | 0.83 | 0.06 | 0.35 | 0.09 | 0.07 | 0.01 | 0.00 | 0.00 |
| VOXILAPREVIR | 0.75 | 0.29 | 0.65 | 0.24 | 0.34 | 0.00 | 0.00 | 0.00 |
| SOFOSBUVIR | 0.70 | 0.16 | 0.86 | 0.03 | 0.00 | 0.23 | 0.00 | 0.00 |
| LEDIPASVIR | 0.67 | 0.53 | 0.86 | 0.15 | 0.01 | 0.00 | 0.00 | 0.00 |
| ELBASVIR | 0.61 | 0.58 | 0.83 | 0.15 | 0.01 | 0.01 | 0.00 | 0.00 |
| GLECAPREVIR | 0.56 | 0.15 | 0.63 | 0.26 | 0.23 | 0.00 | 0.00 | 0.01 |
| VELPATASVIR | 0.54 | 0.68 | 0.81 | 0.16 | 0.02 | 0.00 | 0.00 | 0.01 |
| OMBITASVIR | 0.41 | 0.25 | 0.41 | 0.02 | 0.12 | 0.00 | 0.00 | 0.00 |
| NELARABINE | 0.33 | 0.05 | 0.45 | 0.00 | 0.00 | 0.01 | 0.00 | 0.00 |
| PIBRENTASVIR | 0.20 | 0.32 | 0.35 | 0.02 | 0.01 | 0.00 | 0.00 | 0.01 |
| BAZEDOXIFENE | 0.10 | 0.07 | 0.07 | 0.01 | 0.01 | 0.01 | 0.00 | 0.00 |
| BAZEDOXIFENE ACETATE | 0.10 | 0.07 | 0.07 | 0.01 | 0.01 | 0.01 | 0.00 | 0.00 |
| TEGAFUR | 0.09 | 0.02 | 0.43 | 0.00 | 0.01 | 0.01 | 0.00 | 0.00 |
| MICAFUNGIN | 0.08 | 0.04 | 0.08 | 0.03 | 0.19 | 0.01 | 0.00 | 0.00 |
| MICAFUNGIN SODIUM | 0.08 | 0.04 | 0.08 | 0.03 | 0.19 | 0.01 | 0.00 | 0.00 |
| ACYCLOVIR SODIUM | 0.08 | 0.02 | 0.74 | 0.00 | 0.00 | 0.01 | 0.00 | 0.00 |
| RUCAPARIB CAMSYLATE | 0.08 | 0.09 | 0.10 | 0.03 | 0.00 | 0.01 | 0.00 | 0.01 |
| PERINDOPRIL ERBUMINE | 0.08 | 0.08 | 0.11 | 0.02 | 0.06 | 0.00 | 0.20 | 0.00 |
| DARIDOREXANT HYDROCHLORIDE | 0.08 | 0.03 | 0.07 | 0.03 | 0.00 | 0.00 | 0.00 | 0.01 |
| DARIDOREXANT | 0.08 | 0.03 | 0.07 | 0.03 | 0.00 | 0.00 | 0.00 | 0.01 |
| ACALABRUTINIB | 0.07 | 0.06 | 0.23 | 0.09 | 0.03 | 0.00 | 0.00 | 0.78 |
| PERINDOPRIL ARGININE | 0.07 | 0.07 | 0.11 | 0.02 | 0.06 | 0.00 | 0.15 | 0.00 |
| CYTARABINE | 0.07 | 0.01 | 0.32 | 0.00 | 0.00 | 0.02 | 0.00 | 0.00 |
| PERINDOPRIL | 0.07 | 0.07 | 0.11 | 0.02 | 0.07 | 0.00 | 0.20 | 0.00 |
| IDOXURIDINE | 0.07 | 0.01 | 0.43 | 0.00 | 0.01 | 0.03 | 0.00 | 0.00 |
| MARIBAVIR | 0.07 | 0.04 | 0.20 | 0.00 | 0.00 | 0.01 | 0.00 | 0.00 |
| ACALABRUTINIB MALEATE | 0.07 | 0.05 | 0.24 | 0.09 | 0.02 | 0.00 | 0.00 | 0.78 |
| ADENOSINE PHOSPHATE | 0.07 | 0.03 | 0.17 | 0.00 | 0.00 | 0.02 | 0.00 | 0.00 |
| GANCICLOVIR SODIUM | 0.07 | 0.01 | 0.70 | 0.00 | 0.00 | 0.01 | 0.00 | 0.00 |
| REMDESIVIR | 0.06 | 0.07 | 0.82 | 0.02 | 0.01 | 0.04 | 0.00 | 0.00 |
| FLUDARABINE PHOSPHATE | 0.06 | 0.03 | 0.17 | 0.00 | 0.00 | 0.01 | 0.00 | 0.00 |
| GEMCITABINE | 0.06 | 0.02 | 0.66 | 0.00 | 0.00 | 0.02 | 0.00 | 0.00 |
| GEMCITABINE HYDROCHLORIDE | 0.06 | 0.02 | 0.66 | 0.00 | 0.00 | 0.02 | 0.00 | 0.00 |
| URIDINE TRIACETATE | 0.06 | 0.01 | 0.25 | 0.00 | 0.00 | 0.02 | 0.00 | 0.00 |
| RUCAPARIB | 0.06 | 0.10 | 0.15 | 0.02 | 0.00 | 0.01 | 0.00 | 0.01 |
| PENCICLOVIR | 0.06 | 0.03 | 0.50 | 0.00 | 0.00 | 0.02 | 0.00 | 0.00 |
| AMDINOCILLIN PIVOXIL | 0.05 | 0.01 | 0.04 | 0.01 | 0.02 | 0.01 | 0.00 | 0.00 |
| CAPECITABINE | 0.05 | 0.01 | 0.25 | 0.00 | 0.00 | 0.01 | 0.00 | 0.00 |
| FLOXURIDINE | 0.05 | 0.01 | 0.41 | 0.00 | 0.00 | 0.02 | 0.00 | 0.00 |
| BRIVARACETAM | 0.05 | 0.01 | 0.07 | 0.01 | 0.05 | 0.00 | 0.00 | 0.00 |
| DITHIAZANINE | 0.05 | 0.02 | 0.07 | 0.00 | 0.01 | 0.00 | 0.00 | 0.00 |
| BRINZOLAMIDE | 0.05 | 0.08 | 0.15 | 0.02 | 0.02 | 0.01 | 0.00 | 0.00 |
| ENASIDENIB | 0.04 | 0.04 | 0.05 | 0.00 | 0.00 | 0.00 | 0.00 | 0.00 |
| ANIDULAFUNGIN | 0.04 | 0.05 | 0.12 | 0.01 | 0.22 | 0.00 | 0.00 | 0.00 |

Figure S7: **Top 50 FDA-approved drugs predicted to contain the label 'hcv'.** The Stage-4 approved drugs list from OpenTargets was passed through the CheF label prediction model. Results were sorted by 'hcv' probability. Relevant and high abundance labels displayed for clarity. Green cells represent approved-use labels from on the OpenTargets page, and red cells represent no approved usage relevant to the given term.

