# OpenReview forum: "Mining Patents with Large Language Models Demonstrates Congruence of Functional Labels and Chemical Structures"
_ICLR.cc/2024/Conference — Submitted to ICLR 2024_

### Official Review · Reviewer_HDTT · 2023-10-30

**Soundness:** 3 good
**Presentation:** 3 good
**Contribution:** 2 fair
**Rating:** 3
**Confidence:** 4

**Summary:**

This paper uses ChatGPT as a strong assistant to summarize chemical patent information to provide more functional labels for related molecules. This paper proposes a pipeline containing both the label creation step and the label cleaning step.

**Strengths:**

(1) This reviewer believes the generated large-scale datasets can be a solid contribution to the ai4science community and the label creation process can be transferred to other similar scientific dataset collections;

(2) The label creation and label cleaning process is clearly shown through figure illustrations and paragraph descriptions, which is easy for readers to follow.

**Weaknesses:**

(1) It seems this paper only designs a dataset collection pipeline and no benchmark works are involved. For example, a complete benchmark work should include additional evaluations of some baseline methods. This reviewer thinks only the dataset construction contribution is not sufficient for publishing in ICLR;

(2) Although the problem formulation seems very straightforward (functional label prediction), it still lacks a paragraph to explicitly explain the problem formulation. And need one more paragraph to explain why functional label prediction is an important task (naturally motivated by what application scenarios?).

(3) This reviewer thinks this paper might not be very suitable for the ICLR conference venues (machine learning conference). It is probably more suitable for the domain journal or other conference benchmark tracks if baseline evaluations are further included. This reviewer encourages the authors to make the dataset public. This reviewer thinks probably the most influential dataset format is maintaining the multi-modality, which means including the raw patent text tokens.

**Questions:**

Will the functional label prediction be a simple task for deep learning approaches? This reviewer conjectures that different functional molecules can have very different structures, which might be easy for deep learning approaches to distinguish between them.

---

> ### Author Response · Authors · 2023-11-21
> **Response part 1 to Reviewer HDTT**
>
> We thank you for your comprehensive review, your feedback has allowed us to greatly improve our work. We have made substantial text changes to the story. In the previous manuscript, we over-emphasized the narrative on dataset creation, when this was just a means to perform functional label-guided drug discovery. The revised manuscript has several key changes. First, we emphasize the importance of focusing on chemical function rather than structural nomenclature in the introduction and related work. Second, we introduce the concept of the chemical function landscape and through several analyses demonstrate that it is approximated by the CheF dataset. And third, we now outline the unique advantages that our proposed functional label-guided drug discovery has in comparison to existing text-based chemical models. We believe the reviewer feedback from all parties has greatly increased the quality of our paper, and we want to thank you for your valuable feedback.
>
> Below, we address each of your points in detail.
>
> > It seems this paper only designs a dataset collection pipeline and no benchmark works are involved. [...] This reviewer thinks only the dataset construction contribution is not sufficient for publishing in ICLR. […] This reviewer thinks this paper might not be very suitable for the ICLR conference venues (machine learning conference). It is probably more suitable for the domain journal or other conference benchmark tracks if baseline evaluations are further included.
>
> We understand your concerns on the focus of our paper, and we believe they reflect that we did not properly convey our work in the original submitted manuscript. Previously, we had focused too specifically on the dataset creation, when that was just a means to functional label-guided drug discovery which is, to our knowledge, a novel contribution to the field. To correct this shortcoming, we have overhauled the presented story to better explain the results of our contributions. You will find many text changes to the Abstract, Introduction, Results, and Discussion.
>
> We believe our work serves instead to help build the foundations of functional label-guided drug discovery. First, we introduce the importance of focusing on chemical function, rather than structural nomenclature or miscellaneous terms. This is demonstrated with the creation of our Chemical Function dataset. Then, we introduce the concept of the text-based chemical function landscape, which was found to be highly structured by semantic functional categories (Figure 3). This text-based functional landscape aligned with chemical structural relationships, meaning that it adequately approximated the actual chemical function landscape, implicitly capturing complex physical and biochemical interactions (Figs. 2, 4). We then demonstrate that this functional landscape can be harnessed through a model trained to predict entire chemical functional profiles from structure alone. This model was able to comprehensively assign probabilities to every possible function that a given chemical may have, something that we do not believe has been done prior to our work. This has immense utility for experimentalists (hypothesis generation), pharmaceutical companies (discovery & repurposing), environmental agencies (off-target effects), and electronics manufacturers (photochemical discovery) to name a few.
>
> We do realize that some aspects of technical rigor were not present in our original draft. To remedy this, we include additional supplemental figures analyzing the bias of the dataset (Fig. S3) and a benchmarking of molecular fingerprint-based models on the CheF dataset (Fig. S7).
>
> We hope that you will find the revised manuscript and story satisfactory. We firmly believe that the work presented here is beneficial for the ML chemistry / biochemistry researchers attending ICLR, rather than a general chemistry audience.
>
> > [...] a complete benchmark work should include additional evaluations of some baseline methods.
>
> This is a great suggestion, and the exclusion of such was an oversight on our end as we were focused on the analysis and practical utility of functional label-guided drug discovery rather than dataset creation. To address this, we have added evaluations of several baseline methods on the CheF dataset that may be found in Table S7. From this benchmark process, we found out our previous model was suboptimal. We now use a better model (multi-label logistic regression), and we have redone the relevant downstream analyses.
>
> (continued in next comment)

---

> > ### Author Response · Authors · 2023-11-21
> > **Response part 2 to Reviewer HDTT**
> >
> > > Although the problem formulation seems very straightforward (functional label prediction), it still lacks a paragraph to explicitly explain the problem formulation [...] need one more paragraph to explain why functional label prediction is an important task (naturally motivated by what application scenarios?).
> >
> > This is a good point. Our revised paper now has an explicit problem formulation paragraph at the end of the Introduction and beginning of Results. Additionally, we have spent considerably more time in the Introduction explaining why functional label-guided drug discovery is important. In particular, we discuss pitfalls in structure-based methods predicting whole-body effects and the impact of Gene Ontology to genome understanding and the lack of an equivalent knowledge source for chemicals. We have also better clarified all of the practical applications exhibited in the section Functional label-guided drug discovery. Additionally, we have added more on applications in the discussions: “Models in this paradigm have the potential to automatically annotate chemical function, examine non-obvious features of drugs such as side effects, and down-select candidates for high-throughput screening”.
> >
> > > This reviewer encourages the authors to make the dataset public.
> >
> > We are committed to open science, and believe the best way to increase the development of functional label-guided drug discovery is to have the data accessible for other researchers to build on. The CheF database is already public and may be downloaded at the provided Zenodo link in the Reproducibility Statement section.
> >
> > > This reviewer thinks probably the most influential dataset format is maintaining the multi-modality, which means including the raw patent text tokens.
> >
> > The provided Zenodo link contains code to fetch all of the relevant patent data given a SMILES string / Patent ID that the research community may find useful.
> >
> > However, we believe that high-quality functional labels are at present more powerful than raw patent text. We address this in the rewritten paragraph on text-to-chemical translation in Related Work: “Many existing chemical-to-text translation models have been trained on datasets containing structural nomenclature and irrelevant words mixed with desirable functional information (Edwards et al., 2021; Degtyarenko et al., 2007). Inclusion of structural nomenclature causes inflated prediction metrics for functional annotation or molecular generation tasks, as structure-to-name and name-to-structure is simpler than structure-to-function and function-to-structure. The irrelevant words may cause artifacts during the decoding process depending on the prompt, skewing results in ways irrelevant to the task. In our work, we ensured our model utilized only chemical structure, and not structural nomenclature, when predicting molecular function to avoid data leakage”.
> >
> > > Will the functional label prediction be a simple task for deep learning approaches? This reviewer conjectures that different functional molecules can have very different structures, which might be easy for deep learning approaches to distinguish between them.
> >
> > The task of functional label prediction is actually quite nuanced and poses significant challenges for deep learning approaches. The function of a molecule is determined by its interacting partner (e.g., photons, specific receptors, or the organization of a signaling cascade). This means that models trained to predict functional labels need to approximate the chemical function landscape, a massively high-dimensional space encoding complex interaction relationships. This is addressed several times in the updated manuscript as it is central to our work in the revised story.
> >
> > The diversity and complexity of molecular structures means that molecules with different functions can have subtle and nuanced structural differences, making them challenging to distinguish. And conversely, different molecules can have the same function despite drastically different structure. An example of the former is the drug Thalidomide, which is either therapeutic (prevents morning sickness) or teratogenic (causes birth defects) depending on the stereoisomer (specific handedness of one of the bonds, holding all else constant). And an example of the latter is that there are many distinct mechanisms for a drug to be an antibiotic. This can be seen in beta-lactam antibiotics and tetracyclines both working as antibiotics despite radically different structure and mechanisms. We have added discussion of these points to the manuscript: “This rationale generally holds, but exceptions include stereoisomers with different functions and distinct structures sharing the same function.”
> >
> > We believe that deep learning approaches provide an opportunity to solve the challenging problem of chemical function prediction, but this requires high-quality uncontaminated data with a model fit suited to the problem, both of which we have provided.

---

> ### Comment · Reviewer_HDTT · 2023-11-23
> **Reply to the author's rebuttal**
>
> Dear Author,
>
> Thanks for your rebuttal! After careful reading over the rebuttal, we are mainly concerned about its significance and contribution. This reviewer believes that the functional label prediction task is an important task for real-application scenarios and biological/chemical research. However, the current contribution is not complete enough for an ICLR conference publication, considering this is a representation learning conference although the scope is definitely very wide. We think the best way to prove its significance is to show that some SOTA deep-learning approaches perform worse on the newly proposed benchmark since this is the only way to prove that the new task is a new challenge for current models. However, we only see some traditional and easy baseline approaches in rebuttal (even some easy approaches achieve very decent performance) and we truly think making a comprehensive and effective evaluation is not an easy thing that can be done immediately.
>
> If the key point is more about the data mining techniques, then we think the technical contribution and novelty are probably very limited from a machine learning perspective. The current vision might be more suitable for a domain academic journal.
>
> Therefore, we are sorry to say that we will maintain our score and leave this decision to AC. In general, we acknowledge the solid contribution of this work to the domain research and we believe this work will be finally accepted after rounds of revisions.
>
> Best.

---

### Official Review · Reviewer_Lg3y · 2023-11-01

**Soundness:** 3 good
**Presentation:** 3 good
**Contribution:** 3 good
**Rating:** 8
**Confidence:** 4

**Summary:**

The paper tackles the task of predicting chemical function from chemical structure by mining patents. The proposed methodology uses GPT3.5 to summarise patents and extract functional labels, which are then cleaned and disambiguated by embedding them with a different GPT model and summarizing the clusters. The final labels are then mapped to the corresponding chemical structures of the chemicals associated with the patent. The authors create the Chemical Function (CheF) dataset, containing 100k molecules and the derived functional labels. They show that there is a relationship between the extracted labels and chemical structure of the molecules, by converting the molecules to molecular fingerprints and confirming clusters in the structural space correspond to functional labels when projecting with t-SNE. This finding is further validated by performing a similar analysis on the co-occurrence graph of functional labels. The paper also trains a function prediction model on the molecular fingerprints in the CheF dataset and demonstrates its utility qualitatively on a few real-life examples

**Strengths:**

- **Originality**: One of the core assumptions of the paper is that relationships between chemical structure and function are present in the language itself. The paper proposes an interesting approach to derive functionality from the language itself, rather than chemical structure, and then shows that the learned functionality corresponds to classes of chemical structure
- **Significance**: Overall, the paper combines SOA approaches like LLMs in a novel and interesting way, in order tackle a hard problem (predicting chemical function). Identification of the mechanistic plausible false positives is interesting and could have significant impact in speed up the process of identifying new drugs for drug repurposing
- **Presentation**: The paper has good structure, and claims follow clearly from experiments

**Weaknesses:**

- **Evaluation**: The paper assumes that chemical structures with similar functionality should cluster close to each other in the structural space based on molecular fingerprints; however, this doesn’t necessarily have to be the case - you can have stereoisomers with different functional properties, and you can have chemical compounds with different chemical structure and similar labels. Some of this is visible in the cluster analyses, where molecules belonging to the same functional class are not clustered together. Perhaps a different metric to assess congruence of the two spaces is needed.

**Questions:**

- How is this effort different from recent work on molecular generation from chemical subspaces derived from patents containing specific functional keywords (Subramanian, 2023)?
- What is the classification model used to go from molecular fingerprint to functional label?

---

> ### Author Response · Authors · 2023-11-21
> **Response part 1 to Reviewer Lg3y**
>
> We thank you for your thorough review and insightful feedback on our paper. We appreciate your recognition of the originality and significance of our work and are glad that our approach of functional label-guided drug discovery resonated well with your understanding of the field's challenges. We have made substantial text changes to the story. In the previous manuscript, we over-emphasized the narrative on dataset creation, when this was just a means to perform functional label-guided drug discovery. The revised manuscript has several key changes. First, we emphasize the importance of focusing on chemical function rather than structural nomenclature in the introduction and related work. Second, we introduce the concept of the chemical function landscape and through several analyses demonstrate that it is approximated by the CheF dataset. And third, we now outline the unique advantages that our proposed functional label-guided drug discovery has in comparison to existing text-based chemical models. We believe the reviewer feedback from all parties has greatly increased the quality of our paper, and we want to thank you for your valuable feedback.
>
> Below, we address each of your points in detail.
>
> > The paper assumes that chemical structures with similar functionality should cluster close to each other in the structural space based on molecular fingerprints; however, this doesn’t necessarily have to be the case - you can have stereoisomers with different functional properties, and you can have chemical compounds with different chemical structure and similar labels. Some of this is visible in the cluster analyses, where molecules belonging to the same functional class are not clustered together. Perhaps a different metric to assess congruence of the two spaces is needed.
>
> This is an excellent point. It is indeed true that chemical structures with similar functionalities do not always cluster closely in structural space, and conversely that structurally similar molecules may not always share functionality. An example of the former, as you pointed out, is the label HCV (Hepatitis C Virus) showing distinct clusters depending on mechanism of action (see Figure S3). A well-known example of the latter effect is the drug Thalidomide, which is either therapeutic or teratogenic depending on the stereoisomer.
>
> Despite this, derivatives of active drugs will often maintain their activity provided that the active moieties are maintained. This can be seen across the broad range of beta-lactam antibiotics. It is because of this generally-true property that patent offices allow for the patenting of molecules and their derivatives. And because the CheF dataset was built from patent data, we should expect to see clustering in structure space above random chance. However, we should not expect to see a single cluster for all molecules containing a given label, as some terms have multiple distinct mechanisms (i.e., beta-lactam antibiotics & tetracyclines both act as antibiotics despite different structure). Our metric accounts for this, as it measures distance to the nearest molecule with a shared label rather than measuring number of clusters.
>
> We have further clarified the rationale for our introduced hypothesis, in which we add “Due to molecules and their derivatives being patented together, structurally similar molecules should be annotated with similar patent-derived functions. This rationale generally holds, but exceptions include stereoisomers with different functions and distinct structures sharing the same function”.
>
> > How is this effort different from recent work on molecular generation from chemical subspaces derived from patents containing specific functional keywords (Subramanian, 2023)?
>
> We have rewritten the paragraph comparing our work to Subramanian 2023 to better clarify commonalities and differences. The revised paragraph reads as follows:
>
> “Recent work also focused on molecular generation from chemical subspaces derived from patents containing specific functional keywords, for example, all molecules relating to tyrosine kinase inhibitor activity (Subramanian et al., 2023). This allows for a model that can generate potential tyrosine kinase inhibitors but would need to be retrained to predict molecules of a different functional label. In our work, we focus on label classification rather than molecular generation. Further, we integrate multiple functional labels for any given molecule, allowing us to broadly infer molecular functionality given structure. Generative models could be trained on the described dataset, allowing for label-guided molecular generation without re-training for each label.”
>
> (continued in next comment)

---

> > ### Author Response · Authors · 2023-11-21
> > **Response part 2 to Reviewer Lg3y**
> >
> > > What is the classification model used to go from molecular fingerprint to functional label?
> >
> > In the original paper, we used a feedforward neural network with two hidden layers (dimensions 512, 256 respectively), to output a 1x1,543 vector (size of vocabulary). This was determined using a 5-fold CV.
> >
> > However, we have since benchmarked additional models on the CheF dataset and found that a multi-label logistic regression model on molecular fingerprints actually worked significantly better. We redid the downstream analyses using this updated model. You can find model training details in the Methods under Model Training. We have also clarified mention of which model is being discussed throughout the paper.

---

### Official Review · Reviewer_pFTF · 2023-11-01

**Soundness:** 3 good
**Presentation:** 3 good
**Contribution:** 3 good
**Rating:** 6
**Confidence:** 3

**Summary:**

The paper applies large language models to chemical patents to leverage the information about chemical functionality captured by these resources. Using ChatGPT-assisted patent summarization and word-embedding label cleaning pipeline that paper provides Chemical Function (CheF) dataset, containing 100K molecules and their patent-derived functional labels.

**Strengths:**

The paper is well-written and very clear to its points. The authors define the potential of using language models on chemical patent data, and presents .

**Weaknesses:**

- Detailed figure of the pipeline (may be with a brief example of a patent) may be easier for readers understanding (Figure 1).

**Questions:**

- Can more validations be carried out by searching over the pubmed?
- Would it be possible to employ the connection of chemicals with genes or diseases to extend the usage in drug repositioning or drug combinations?
- In using LLMs, were there any bias in the generated summarizations?
- What was the reason for using the Tanimoto similarity?
- How long (in terms of time) does the whole framework take?

---

> ### Author Response · Authors · 2023-11-21
> **Response to Reviewer pFTF**
>
> Thank you very much for your review, your feedback has allowed us to greatly improve our work. We have made substantial text changes to the story. In the previous manuscript, we over-emphasized the narrative on dataset creation, when this was just a means to perform functional label-guided drug discovery. The revised manuscript has several key changes. First, we emphasize the importance of focusing on chemical function rather than structural nomenclature in the introduction and related work. Second, we introduce the concept of the chemical function landscape and through several analyses demonstrate that it is approximated by the CheF dataset. And third, we now outline the unique advantages that our proposed functional label-guided drug discovery has in comparison to existing text-based chemical models. We believe the reviewer feedback from all parties has greatly increased the quality of our paper, and we want to thank you for your valuable feedback.
>
> Below, we address each of your points in detail.
>
> > The paper is well-written and very clear to its points. The authors define the potential of using language models on chemical patent data, and presents .
>
> We are glad to hear that the clarity of our paper met your expectations, however, did this response get cut off? It appears the sentence stopped short.
>
> > Detailed figure of the pipeline (may be with a brief example of a patent) may be easier for readers understanding (Figure 1)
>
> We recognize the importance of visual aids for those unfamiliar with patents. To aid this, we have added a figure outlining the summarization process for a specific example which may be found in Figure S2.
>
> > Can more validations be carried out by searching over the pubmed?
>
> In our paper, we use PubMed articles to confirm our model-derived hypothesis of the Hepatitis C Virus antiviral’s mechanism of action (Section 3.4). If you were instead referring to the idea that we should expand our data to PubMed-associated molecules, we completely agree and thank you for the suggestion. To note this in the paper, we have added a sentence to the discussion that reads “Additionally, by restricting the dataset to chemicals with <10 patents, it neglects important well-studied molecules like Penicillin. The inclusion of over-patented chemicals could be accomplished by using only the most abundant k terms for a given molecule, using a fine-tuned LLM to only summarize patents relevant to molecular function (ignoring irrelevant patents on applications like medical devices), or employing other data sources like PubChem or PubMed to fill in these gaps”. We aim to pursue this idea in future work.
>
> > Would it be possible to employ the connection of chemicals with genes or diseases to extend the usage in drug repositioning or drug combinations?
>
> This is an excellent suggestion, and it is exactly where we are headed in continuations of this project. We believe these relationships can be unified with our CheF dataset, allowing for the discovery of new chemical-gene and chemical-disease relationships via commonalities in patent word space. To note this in the paper, we have added a sentence to the discussion that reads “Further quality increases may result from integration of well-documented chemical-gene and chemical-disease relationships into CheF”.
>
> > In using LLMs, were there any bias in the generated summarizations?
>
> There were no obvious biases in the generated summarizations. To confirm this, we analyze the distribution of the generated (uncleaned) labels based on abundance. These appear to reflect underlying biases in the patent literature, and more broadly pharmaceutical incentives, rather than from the LLM. This analysis may be found in Figure S3.
>
> > What was the reason for using the Tanimoto similarity?
>
> Tanimoto similarity is the standard method to compute similarity between binary molecular fingerprint vectors, see Dávid Bajusz (2015). This allows us to compute similarity whether or not the molecule contains a particular substructure. If we had been using float-valued vector representations (i.e., learned molecular embeddings), we would have opted for a more fitting similarity metric such as cosine similarity or Euclidean distance. To clarify this, we have added a citation of the metric at its first mention.
>
> > How long (in terms of time) does the whole framework take?
>
> The framework is largely limited by the API calls to retrieve a patent and summarize it with ChatGPT. We report the results of this in the Methods section: “Fetching patent information and summarizing with ChatGPT, this method’s bottleneck, took 6 seconds per molecule with 16 CPUs in parallel. This could be sped up to 3.9 seconds by summarizing per-patent rather than per-molecule to avoid redundant summarizations, and even further to 2.6 seconds by using only US and WO patents”.

---

### Official Review · Reviewer_qEJ1 · 2023-11-01

**Soundness:** 2 fair
**Presentation:** 3 good
**Contribution:** 2 fair
**Rating:** 5
**Confidence:** 4

**Summary:**

This paper applies large language models for the task of identifying and providing summaries of the functional labels associated with chemical molecules from chemical patents. Through their efforts, the authors introduce the CheF dataset and employ label embedding and clustering techniques to obtain cleaned functionality labels. The paper then offers a comprehensive analysis of the generated dataset.

**Strengths:**

1.	The research addresses a unique chemical problem by introducing an innovative method for extracting molecule functionality information using ChatGPT. The introduction of the CheF dataset is a commendable contribution that has the potential to benefit the broader chemistry community.
2.	The analysis of the CheF dataset, including  the relationships between the functional labels and the chemical structure space as well as the label co-occurrences, provides valuable insights that enhance readers' comprehension of the dataset.

**Weaknesses:**

1.	From a technological perspective, the paper's contribution appears somewhat restrained. The use of ChatGPT for text summarization is not a novel concept. As such, the manuscript might find a more fitting audience in journals or conferences with a chemistry focus.
2.	There are concerns regarding the accuracy of labels generated by ChatGPT. While 98.2% of the labels were found valid, solutions for addressing the remaining 1.8% are not discussed. Such inaccuracies could introduce noise into the CheF dataset.
3.	The selection criterion that omits molecules associated with more than 10 patents suggests the dataset may be missing data on prevalent molecules. The absence of these molecules might limit the dataset's reach and implications.
4.	A comparative analysis with other established molecule-text datasets like ChEBI and PubChem would be beneficial. Additionally, the paper could emphasize the practical applications of the CheF dataset, especially its potential role in drug discovery, to underscore its unique advantages.

**Questions:**

1.	In the Section 3.1 (FUNCTIONAL LABELS MAP TO NATURAL CLUSTERS IN CHEMICAL STRUCTURE SPACE) part, the (a) (e), and (g)  don't appear to distinctly exhibit clustering. Are there more significant examples of other labels?

---

> ### Author Response · Authors · 2023-11-21
> **Response part 1**
>
> Thank you very much for your review, your feedback has allowed us to greatly improve our work. We have made substantial text changes to the story. In the previous manuscript, we over-emphasized the narrative on dataset creation, when this was just a means to perform functional label-guided drug discovery. The revised manuscript has several key changes. First, we emphasize the importance of focusing on chemical function rather than structural nomenclature in the introduction and related work. Second, we introduce the concept of the chemical function landscape and through several analyses demonstrate that it is approximated by the CheF dataset. And third, we now outline the unique advantages that our proposed functional label-guided drug discovery has in comparison to existing text-based chemical models. We believe the reviewer feedback from all parties has greatly increased the quality of our paper, and we want to thank you for your valuable feedback.
>
> Below, we address each of your points in detail.
>
> > From a technological perspective, the paper's contribution appears somewhat restrained. The use of ChatGPT for text summarization is not a novel concept. As such, the manuscript might find a more fitting audience in journals or conferences with a chemistry focus.
>
> We acknowledge your concerns on the scope of our paper and believe it means that we did not properly convey our work in the original submitted manuscript. Previously, we had focused too specifically on the dataset creation, when that was just a means to functional label-guided drug discovery which is, to our knowledge, a novel contribution to the field. To correct this, we have edited each section to better relate our contributions.
>
> We believe our work helps build the foundations of functional label-guided drug discovery. First, we introduce the importance of focusing on chemical function, rather than structural nomenclature or miscellaneous terms. This is demonstrated with the creation of our Chemical Function dataset. Then, we introduce the concept of the text-based chemical function landscape, which was found to be highly structured by semantic functional categories (Figure 3). This text-based functional landscape aligned with chemical structural relationships, meaning that it adequately approximated the actual chemical function landscape, implicitly capturing complex physical and biochemical interactions (Figs. 2, 4). We then demonstrate that this functional landscape can be harnessed through a model trained to predict entire chemical functional profiles from structure alone. This model was able to comprehensively assign probabilities to every possible function that a given chemical may have, something that we do not believe has been done prior to our work. This has utility for experimentalists (hypothesis generation), pharmaceutical companies (discovery and repurposing), environmental agencies (off-target effects), and electronics manufacturers (photochemical discovery) to name a few.
>
> We do realize that some aspects of technical rigor were not present in our original draft. To remedy this, we now include additional supplemental figures analyzing the bias of the dataset (Fig. S3) and a benchmarking of molecular fingerprint-based models on the CheF dataset (Fig. S7).
>
> We hope that you will find the revised manuscript satisfactory. We believe that the work is beneficial for the ML chemistry / biochemistry researchers attending ICLR, rather than a general chemistry audience.
>
> > There are concerns regarding the accuracy of labels generated by ChatGPT. While 98.2% of the labels were found valid, solutions for addressing the remaining 1.8% are not discussed. Such inaccuracies could introduce noise into the CheF dataset.
>
> We agree that despite the high base accuracy of 98.2%, addressing the 1.8% of inaccuracies is indeed important. It is likely that fine-tuning an LLM on high-quality patent-extract functional data will increase this accuracy even higher, thereby enhancing the overall quality and reliability of the CheF dataset. A sentence has been added to the discussion to note this: “Increasing label quality and ignoring extraneous claims might be achieved through an LLM fine-tuned on high-quality examples”.
>
> (continued in next comment)

---

> > ### Author Response · Authors · 2023-11-21
> > **Response part 2**
> >
> > > The selection criterion that omits molecules associated with more than 10 patents suggests the dataset may be missing data on prevalent molecules. The absence of these molecules might limit the dataset's reach and implications.
> >
> > We acknowledge the concern for potential limitations that the patent selection criterion imposes. The selection criterion was designed to maintain dataset quality, and we believe it is not as restrictive as it first appears. The SureChEMBL database we used contains 32,005,106 molecules, and our criterion reduces this to 28,207,428 molecules, only a 12% decrease (these numbers have been added to Results and Methods). This means that a high proportion of chemical space is still included, but this does not solve the problem of omitting the well-studied molecules. This is addressed in the discussion as follows: “The inclusion of over-patented chemicals could be accomplished by using only the most abundant k terms for a given molecule, using a fine-tuned LLM to only summarize patents relevant to molecular function (ignoring irrelevant patents on applications like medical devices), or employing other data sources like PubChem or PubMed to fill in these gaps.”
> >
> > > A comparative analysis with other established molecule-text datasets like ChEBI and PubChem would be beneficial.
> >
> > We agree that a comparative analysis with similar datasets would add value to the paper, and thank the reviewer for the good suggestion. This comparison can be found in Table S3 of the revised manuscript. This table highlights the CheF dataset’s unique aspects in comparison to existing chemical function datasets.
> >
> > > Additionally, the paper could emphasize the practical applications of the CheF dataset, especially its potential role in drug discovery, to underscore its unique advantages.
> >
> > We agree that emphasizing practical applications is important in comparison to existing methods. To better emphasize this point, we have made several changes to the paper. In the introduction, we now contrast our text-based paradigm to existing structure-based drug discovery methods, noting that our methodology utilizes the massive corpus of chemical literature. Secondly, we have clarified and re-written the section Functional label-guided drug discovery. In this section, we outline the unique ability of models in our proposed paradigm, using the CheF dataset, in several practical drug discovery roles: antiviral mechanism discovery, identifying potentially novel serotonin drugs, and discovering possible hepatitis C virus antivirals that could be repurposed from existing FDA approved drugs. We hope that this clarifies the practical applications of our work.
> >
> > > In the Section 3.1 (FUNCTIONAL LABELS MAP TO NATURAL CLUSTERS IN CHEMICAL STRUCTURE SPACE) part, the (a) (e), and (g) don't appear to distinctly exhibit clustering. Are there more significant examples of other labels?
> >
> > We acknowledge that Figures 2(a), 2(e), and 2(g) do not visibly exhibit strong clustering. However, the degree of clustering was measured, and it is shown in Figures 2(b), 2(d), 2(f), & 2(h), where each example was found to have significant clustering compared to the background. We believe that lack of visual clustering is an artifact of the t-SNE projection on molecular fingerprint vectors. A closer examination of the clusters in Figure 2(a) is shown in Figure S3, in which we found distinct modes of action for the various clusters of molecules containing the label HCV (Hepatitis C Virus).
> >
> > To help aid the reader, we have added additional examples of strongly visually clustering labels which may be found in Figure S5. However, we believe that keeping HCV, 5-HT, and Serotonin cluster examples in the main text helps with consistency as analyses on these labels occur throughout the paper. Further cluster visualization may be found at https://chefdb.app/.

---

### Meta-Review · Area_Chair_wUjr · 2023-12-06

**Metareview:**

This work proposes adopting GPT3.5 to summarise chemical patents, extract functional labels, and clean the labels by embedding them with a different GPT model and summarizing the clusters. The authors make use of the powerful ability of LLM and define the potential of using language models on chemical patent data, and present. However, the reviewers also point out the limited technological contributions, and this work can be regarded as an application of LLM in the chemistry area. Considering the mixed opinions, I lean towards rejection.

**Justification For Why Not Higher Score:**

Please see the meta-review.

**Justification For Why Not Lower Score:**

N/A

---

### Decision · Program_Chairs · 2024-01-16

Reject